# Position: Humans Co-exist, So Must Embodied Artificial Agents.

## Abstract

Modern embodied artificial agents excel in static, predefined tasks but fall short in dynamic and long-term interactions with humans. On the other hand, humans can adapt and evolve continuously, exploiting the situated knowledge embedded in their environment and other agents, thus contributing to meaningful interactions. This position paper introduces the concept of co-existence for embodied artificial agents and argues that it is a prerequisite for meaningful, long-term interaction with humans. We take inspiration from biology and design theory to understand how human and non-human organisms foster entities that co-exist within their specific niches. Finally, we propose key research directions for the machine learning community to foster co-existing embodied agents, focusing on the principles, hardware and learning methods responsible for shaping them.

## 1. Introduction

Modern artificial intelligence systems have shown remarkable performance across diverse tasks such as the high-quality generation of data (image, text, video) (Ho et al., 2020; Achiam et al., 2023; Lu et al., 2023), the creation of interactive world models (Bruce et al., 2024; Alonso et al., 2024), and outperforming humans in complex decision-making tasks (Silver et al., 2016; Vinyals et al., 2019; Vasco et al., 2024). Fundamentally, three ingredients have been mostly responsible for this recent surge in performance: the creation of large-scale models (Vaswani et al., 2023; Dosovitskiy et al., 2021), the curation (or creation) of internet-scale datasets (Schuhmann et al., 2022; Hebart et al., 2023) and a computationally-intensive offline training process (Radford et al., 2021; Zhai et al., 2022; Brown et al., 2020). This recipe has also been replicated for real-world robotic systems, resulting in the creation of large-scale datasets of expert-level interaction data in the real-world (O'Neill et al., 2023) and in simulation environments (Wang et al., 2023). This approach has led to progresses in learning generalist robotic policies, able to perform a wide variety of manipulation and navigation tasks (Black et al., 2024; Zeng et al., 2024).

As a community, we now envision concrete use cases of *embodied artificial agents*[1] for human interaction. Despite their remarkable progress in controlled environments (Brohan et al., 2023), embodied agents still struggle to gain a foothold in real-world, in-the-wild, scenarios (Auger, 2022). Rodney Brooks' famous quip, "The world is its own best model" is often used to encapsulate the problem of conceiving and deploying embodies artificial agents in the real world (Bharadhwaj, 2024). However, we highlight that this challenge does not emerge only from the complex and dynamic nature of the real world (which makes the optimization problem dynamic as well): it also emerges from the fact that the real world is *constantly being viewed* as an optimization problem (Stanley & Lehman, 2015). Interaction in-the-wild instead is co-constructed with the humans in-the-wild (Frauenberger, 2019), which is at odds with the dominant *problematize-solve-optimize-deploy* workflow of the machine learning community (Jordan et al., 2024).

**We argue that our current approach to agent design is unsuitable for meaningful long-term interaction with humans.** In Section 2, we discuss why current embodied artificial agents are unable to cope with the strong dynamic nature of human interaction and their inability to participate in its ongoing evolution. **We emphasize the need for a new paradigm for *co-existing* embodied agents**: systems capable of continuously leveraging the diverse and situated knowledge of both the user and the environment, highlighted in Figure 1, to establish meaningful and reciprocal interactions with the elements of its system. In Section 3, we provide a formal definition of co-existence, meaning, and reciprocity in the context of embodied artificial agents. In Section 4 we look to biology and design theory, two fields that are epistemically grounded in the real world, to

---
[1]Anonymous Institution, Anonymous City, Anonymous Region, Anonymous Country. Correspondence to: Anonymous Author <anon.email@domain.com>.

Preliminary work. Under review by the International Conference on Machine Learning (ICML). Do not distribute.

---
[1]We follow Paolo et al. (2024) that defines embodied artificial agents as "agents that interact with their physical environment, emphasizing sensorimotor coupling and situated intelligence". Throughout this paper, we use the terms *agent*, *embodied agent*, and *embodied artificial agent* interchangeably for simplicity.

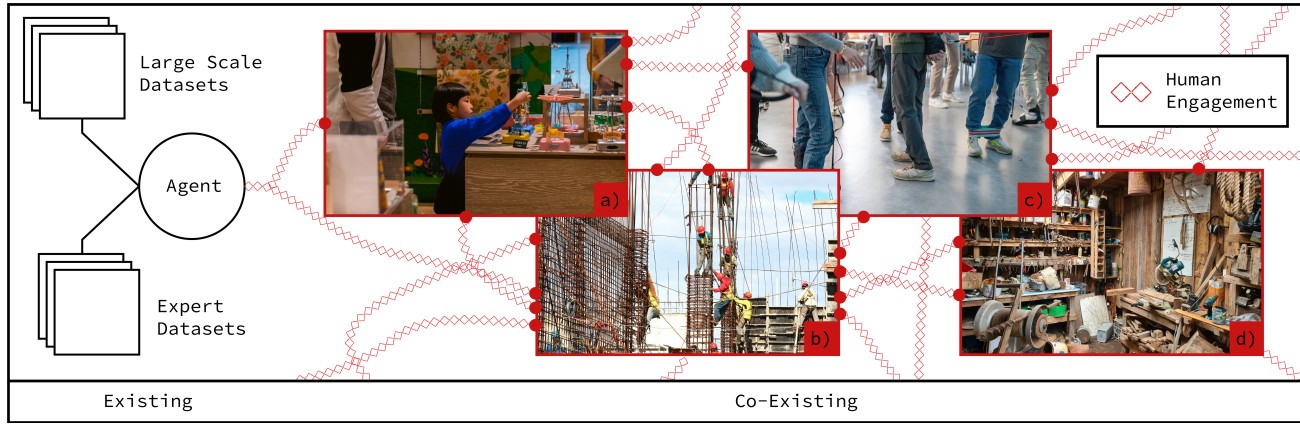

*Figure 1.* Current agents exist in the real-world, leveraging knowledge obtained from large-scale datasets and specific expert-level datasets to interact. We argue that embodied artificial agents must not only adapt to scenarios such as the ones pictured above but participate in their continual evolution. To do so, they must *co-exist*: be able to establish meaningful and reciprocal interactions with the user and its particular environment by leveraging their diverse and situated knowledge. To this end, agents should engage the end user as a designer, i.e., the *connoisseur* of their own situation. We depict four environments where humans possess situated knowledge that lies outside of the scope of large-scale and expert datasets: a) a toy store; b) an exploratory dance class; c) a construction site; d) a messy workshop.

understand how co-existence might look in the context of embodied agents [2]. We showcase how biological organisms leverage properties of the real world to take form during development (converge), and evolve in times of environmental changes (diverge). Similarly, we explore the concept of the *double diamond* (Sharp et al., 2023), and explain how this convergent and divergent process can be envision as a framework to how humans interact with embodied artificial agents in the future.

In Section 5 we discuss alternative viewpoints to co-existence and in Section 6 **we highlight key research directions for the machine learning community to develop co-existing agents**. We focus on the learning methods that enable co-existence, the physical subtract that sustains it, and the principles responsible for shaping it. Additionally, we discuss the ethical considerations in designing embodied agents that co-evolve and play a role in shaping the future of human interactions. We hope that the ideas in this work serve as a bridge, enabling the machine learning community to actively engage with the design research community in forging a path toward co-existing embodied agents.

## 2. Current Embodied Agents *Exist*

Recent advancements in perception, learning and hardware systems have enabled embodied agents to successfully perform complex actions in unstructured environments (Ho et al., 2020; Achiam et al., 2023; Lu et al., 2023). We praise these advancements and believe that the current paradigm

---

[2]Our position builds on past parallels between computers and biological processes (Winograd & Flores, 1986; Brooks, 1991; Clark, 2001) towards an everyday reality with such agents.

(based on multimodal foundation models for perception, reasoning and interaction) is sufficient for these agents to *exist* with humans and their environment.

However, we argue that the disregard of the issues pertaining to current embodied agents can have technical and cultural repercussions if employed widespread in our societies. In particular, we focus on two fundamental properties of these agents: their *stagnant* nature, a consequence of having their abilities fixed at a specific moment in time, and their *generic* nature, due to their instantiation based solely on large amounts of data. As current embodied agents are stagnant and generic, their widespread adoption risks conditioning the evolution of their interactions towards overly homogeneous ones, a phenomenon we denote by *steamrolling*.

### 2.1. Current embodied agents are stagnant

Currently, to train embodied agents we implicitly assume that there exists a *predefined* underlying data distribution (e.g., over the sentences people use when feeling happy, or over the possible socially accepted distances from humans while navigating a crowded room), from which we can extract representative examples to train and evaluate the agent. Furthermore, it is assumed that this data distribution is *static* in time. As such, most of the knowledge acquisition and behavior exploration by the agent happens before it is deployed in a specific environment[3]. This inability to deal

---

[3]Some recent approaches for training embodied AI agents mimic the generative pretraining of language models and additionally use a fine-tuning phase to adapt the overall behavior of the agent to a specific task. However, we note that the fine-tuning data distribution is also itself predefined and static.

with changes in their own knowledge and their environment leads to their *stagnant* nature.

Humans will adapt and change their behavior according to the environment they are situated in, but also participate in its shaping (Dourish, 2001). A classic example can be found in medical record cards in hospital beds: Nygren & Henriksson (1992) found that the physical properties of the card (e.g., the handwriting, wear, tear, other marks) were contributing to the physician's decisions pertaining to both the patient and the activities surrounding their care. The hospital's culture and workflows are not converging to a "fixed" version, rather, they are perpetually evolving as the people, the environment and their interactions change. This is not only happening on a high functioning level: Vergunst & Ingold show that even lower level motor skills, such as the way humans walk, are highly socialized and both culturally and contextually dependent. Therefore, a stagnant agent placed in this system would not be able to participate in this mutual shaping, as its behavior is a function of knowledge of a fixed point in time, which can be outdated at deployment time. Even a well-adapted agent at deployment will drift from the culture as the system evolves.

### 2.2. Current embodied agents are generic

Recent advances in machine learning have focused on extracting broad patterns (e.g., grammar and social norms) from large-scale data to bootstrap the behavior of embodied agents (Szot et al., 2023; Yuan et al., 2024). While learning general rules is valuable, we emphasize the crucial distinction between being generic and being general. *General* knowledge captures fundamental principles that apply broadly across most, if not all, cases. In contrast, *generic* knowledge is applied across many situations without accounting for their specific nuances or contextual diversity. By learning generic information from large-scale datasets, agents reinforce (potentially harmful) biases that exist on such data (Parreira et al., 2023): for example, image generation models produce images of white men for the prompt *"a software engineer"* and women with darker skin tone for the prompt *"a housekeeper"* (Bianchi et al., 2023). Current embodied agents, which often employ these large-scale models for interaction purposes, also rely on generic knowledge. Exploiting only generic knowledge is also inefficient. For example, compare a highly controlled space such as a factory, where workbenches and machines are specifically configured, to a home or office space. Each instance of a home or office is unique and contains *situated knowledge* that is specific to its configuration and the humans in it (Dourish, 2001). An agent that relies solely on generic knowledge, is at a clear disadvantage against an agent that also exploits situated knowledge and, just as importantly, contributes to the ongoing exploration and exploitation of culture and workflow in the space (Gillet et al., 2024).

### 2.3. Current embodied agents steamroll

The nature of current embodied agents poses technical, practical, and moral risks. When a *stagnant* and *generic* agent is placed in a dynamic environment, surrounded by adaptive agents such as humans, then it is the adaptive elements that change. This means that the non-adopting agent is not participating in the continually changing dynamic environment. With widespread adoption, the cultures and workflows of these environments begin to converge towards the ones dictated by the stagnant and generic agents. We denote this phenomenon by *steamrolling*.

This phenomenon can already be observed in the recent use of large language models (LLMs) for text generation: Geng & Trotta estimates that 35% of all scientific paper abstracts in computer science are now written in "LLM-style". "Language does not mirror the social; it also helps to create it" writes Coeckelbergh. In the context of embodied artificial agents, we expect steamrolling to inhibit the divergent, evolution of behavior in each particular environment, in favor of reinforcing existing behavior.

Steamrolling impacts not only human behavior but also the future capabilities of the agents we develop: a model trained on a progressively narrower distribution (such as data curated from its own outputs) suffers from rapid degradation in the quality of its generated output (Shumailov et al., 2024).

## 3. Future Embodied Agents Must *Co-exist*

### 3.1. Definition of co-existence

Long-term interactions between humans and embodied artificial agents have been extensively studied by the robotics community (Leite et al., 2013; de Graaf et al., 2016; Laban et al., 2024), focusing on specific properties of the interaction such as acceptance (de Graaf et al., 2016), engagement (Rakhymbayeva et al., 2021; Leite et al., 2014) and disclosure (Naneva et al., 2020; Ligthart et al., 2019). Here we take a holistic view of the long-term interactions of embodied agents within a system and provide a general-purpose, formal definition of co-existence.

> **Definition**: An embodied artificial agent is *co-existing* in a system if it sustains meaningful and reciprocal interactions with humans and their environment over time.

Consider a system $S = \{A, H, E\}$ consisting of an embodied agent $A_t$ present in a specific environment $E_t$ alongside a human user $H_t$, at a given time $t$. There exists a quality function $Q_O(t)$ that overall describes the system and its evolution, measured from the point of view of an observer

$O \in S$ [4]. The quality function is influenced by the interactions between the agent, user and the environment. We note that the goal of the agent does not necessarily align with this quality function as it may be independent of its intended task (e.g., a household robot assisting with chores may perform its tasks efficiently but disrupt the human's workflow and create frustration).

We can define two categories of interactions within this system. A unilateral interaction $X_t \rightarrow Y_t$ occurs if the state of element $Y$ of the system at the next time step $(t+1)$ is influenced by element $X$, while the next state of $X$ remains independent of $Y$,

$$Y_{t+1} = f_Y(Y_t, X_t, y_t, x_t), \quad X_{t+1} = f_X(X_t, x_t), \quad (1)$$

where $f_X, f_Y$ are unknown and dynamic transition functions, and $x_t, y_t$ are the actions of $X$ and $Y$ at time $t$. Similarly a *reciprocal* interaction $X_t \leftrightarrow Y_t$ occurs if the next state of both elements are mutually influenced,

$$Y_{t+1} = f_Y(Y_t, X_t, y_t, x_t), \quad (2)$$
$$X_{t+1} = f_X(X_t, Y_t, x_t, y_t). \quad (3)$$

Interactions influence the long-term quality of the system, which can be measured after a (system-dependent) time horizon threshold $T_S$. We define a *meaningful* interaction as one that, given sufficient time (i.e., in the long-term), does not decrease the overall quality of the system, as evaluated by all elements of the interaction, compared to the absence of such interaction. Formally,

$$\exists T_S > t, \forall t' > T_S, \forall O \in \{X, Y\}:$$
$$Q_O(t' \mid X_t \rightarrow Y_t) \geq Q_O(t' \mid \emptyset), \quad (4)$$

where $\emptyset$ denotes no interaction and the conditional quality function $Q_O(t' \mid X_t)$ indicates the value of the quality function at $t'$ given that process $X$ occurred at $t < t'$. A *co-existing* agent $A^*$ is then defined as an agent able to maintain reciprocal and meaningful interactions in the long term. Intuitively, this means that, in the long run, the agent benefits the system more than its removal would. Formally,

$$\exists T_S > t, \forall t' > T_S, \forall O \in \{A^*, H\}: \quad (5)$$
$$Q_O(t' \mid A_t^* \leftrightarrow (H_t, E_t), H_t \leftrightarrow E_t) \geq Q_O(t' \mid H_t \leftrightarrow E_t).$$

In Appendix A we present additional considerations and discuss the limitations of our formulation of co-existence, such as the existence of a single human user and the closed nature of the system. In Appendix B, we examine whether current embodied agents already co-exist and provide examples illustrating why they fall short.

---

[4]We do not provide a concrete instantiation of the quality function as it is system-dependent: it can encapsulate several properties of human-robot interaction, some previously enumerated.

### 3.2. Properties of Co-existing Embodied Agents

**Situated** A co-existing agent $A^*$ should actively leverage the fact that it is situated within a specific environment. Rather than relying solely on pretrained knowledge, the agent should leverage the unique situated knowledge embedded in the user and their environment. This capability reflects the agent's *speciation* to its particular system. Formally, this can be expressed as:

$$\exists T_S > t, \forall t' > T_S, \quad (6)$$
$$\forall O \in \{A^*, H\}, \forall O' \in \{A^*, H'\}: \quad (7)$$
$$Q_O(t' \mid A_t^* \leftrightarrow (H_t, E_t)) > Q'_{O'}(t' \mid A_t^* \leftrightarrow (H'_t, E'_t)),$$

where we define a distinct system $S' = \{A^*, E', H'\}$ with its own quality function $Q'_{O'}(t)$, but involving the same agent. Note that, contrary to the generic nature of current embodied agents, we argue that the behavior of co-existing agents should be such that it improves the quality of their specific system, even if the same behavior would result in a *overall quality decrease* in other distinct systems.

**Mutable** A co-existing agent $A^*$ should be capable of continuously adapting its behavior while also influencing the behavior of other elements within the system. Formally, this adaptability relates to the concept of reciprocal interactions:

$$\exists T_S > t, \forall t' > T_S, \forall O \in \{A^*, H\}: \quad (8)$$
$$Q_O(t' \mid (H_t, E_t) \leftrightarrow A_t^*) > Q_O(t' \mid (H'_t, E'_t) \rightarrow A_t^*).$$

This condition implies that co-existing agents and humans should be able to mutually shape each other in ways that enhance the overall quality of the system. In contrast, the *stagnant* nature of current embodied systems often necessitates forcefully adapting the human user (through training) or modifying the environment to fit the agent.

Importantly, changes in the agent's behavior do not always lead to an *immediate* improvement in system quality and may sometimes have the opposite effect. As discussed in Section 4, co-existing agents must be capable of generating divergent behavior even within a closed system. This ability is crucial for the long-term success of the system as it enables the exploration of alternative solutions, not only in the agent's behavior but also in how its behavior impacts the other elements of the system.

## 4. Co-existence Elsewhere

The challenge of developing artificial agents that co-exist remains an open question. However, both biological and human-designed systems offer valuable insights, having produced entities that successfully co-adapt and support meaningful interactions within their respective niches. In this section, we explore research in biology and design that

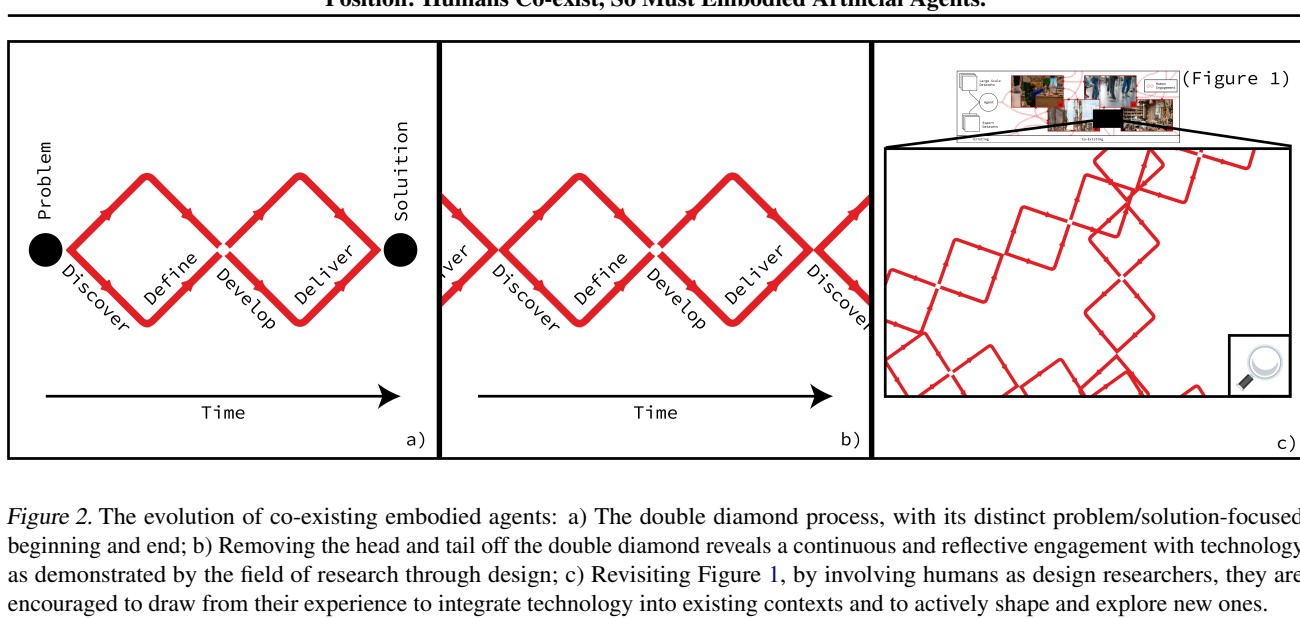

*Figure 2.* The evolution of co-existing embodied agents: a) The double diamond process, with its distinct problem/solution-focused beginning and end; b) Removing the head and tail off the double diamond reveals a continuous and reflective engagement with technology as demonstrated by the field of research through design; c) Revisiting Figure 1, by involving humans as design researchers, they are encouraged to draw from their experience to integrate technology into existing contexts and to actively shape and explore new ones.

highlights the value of mutability and situated knowledge in fostering co-existence[5].

### 4.1. Co-existence in Biology

Biological systems offer a unique perspective on co-existence, showing how living organisms evolve, adapt, and sustain themselves in their own niches. Unlike current embodied agents, which assume that all necessary knowledge can be extracted from data and encoded, biology balances encoded information with meaningful interactions with the physical world to shape adaptation and survival. In this section, we present examples from genetics and developmental biology that explore how biology navigates this balance.

**Not everything is in the genome**   Underlying the majority of machine learning models is the assumption that all necessary knowledge to act/decide optimally can be extracted from data and subsequently exploited. However, biology provides us with a perspective shift in regards to the nature and role of data in the evolution of agents. To illustrate how encoded information is only one part of what shapes biological organisms, we turn to the Human Genome Project (Collins & Fink, 1995). When this project successfully sequenced the entire human genome it was widely believed that the genome could define what humans are, a "instruction book for life". However as Ball explains, the project instead marked the beginning of a paradigm shift in biology that de-throned the genome as an encrypted source of life's secrets. Instead it was shown that an organism is not only defined by the genome but also by principles of self-organization that are enacted by being situated in the

physical world (Ball, 2023).

A striking example of this new reality can be seen in developmental biology, where the number, thickness, and size of a rodent's digits were not found to be encoded in the genome. Instead, a timing of particular proteins (namely BMP, SOX9 and WNT) that disperse in physical space determines the number of digits and the space between them (Raspopovic et al., 2014). Raspopovic et al. discovered that they could manipulate the activity of these proteins and could thus influence the number of digits formed and their thickness. This example shows how the characteristics of the physical world play a role in defining information and intelligence[6], providing an extremely efficient way of acting in the world (Ball, 2023).

**Biology is not an optimizer**   Leveraging the physical world is not only about converging on optimally efficient solutions but also about diverging from locally competitive landscapes. It is a common misconception that biology is an optimizer. As Stanley & Lehman write: "Early evolutionists believed, and indeed many non experts still believe, that evolution is progressive, moving towards some sort of objective perfection, a kind of search for the über organism". In fact, "most evolutionary changes at the molecular level [DNA] are caused not by Darwinian selection but by random drift of mutated genes that are selectively neutral" (Yahara, 1999). As an example, let us consider the protein HSP90, where HSP denotes for "heat shock protein". HSP90 was discovered to have a kind of plasticity modulation effect on the

---

[5]Our focus on biology and design is intentional: these fields are rooted in practice-based epistemologies, emphasizing creation and interaction over mere evaluation.

[6]These morphogenetic patterning processes are not only commonplace in biology but to the physical world at large. In 1952 Alan Turing published a mathematical model that predicted this process (Turing, 1952), over time this mechanism was found to account for phenomena outside of biology, including windswept sand and solidified alloys and ant behavior (Ball, 2023).

body plans of the common fruit fly. In warmer conditions, this protein enables more variation in the morphology of the fruit flies, in places such as its abdomen, bristles, eyes, legs, thorax and wings (Rutherford & Lindquist, 1998). In addition, these traits were able to be passed down immediately to the next generation (Yahara, 1999). It is argued that processes like the ones observed here played a large part in periods of intense diversification in living organisms during the Cambrian explosion (Ball, 2023). This alludes to the idea that evolution, whilst highly divergent, is both *bound* and *liberated* by the laws of nature: by using existing building blocks in creative ways, it is able to keep a tension between convergence and divergence (Gerhart & Kirschner, 2007), conditioning and stimulating exploration and exploitation of novel solutions within its own laws.

### 4.2. Co-existence in Design

We have seen how biological organisms exploit being situated in the world to balance convergence and divergence in order to foster coexistence in their physical setting. However, how a human could instantiate a similar process, with their plans, goals, morals, and aesthetics is still unclear. The answer lies in the divergent and convergent *processes* of design which cause an individual to *engage reciprocally* with technology and its environment, as we highlight in Figure 2.

**The double diamond** The design process often converges to a design outcome, due to performance specifications (Cross, 2000), or intended functions or styles (Rodgers, 2011). In order to deliver an outcome, methods and heuristics exist within each design discipline (Tomitsch et al., 2020; Cross, 2000). But beneath these formalizations lies a practice that is tacit and with an improvisational dimension. This dimension is not only a function of expert knowledge from formal education (industrial, mechanical, electrical, graphical, architectural, etc.), but a craft-like knowledge of their materials, and a situated understanding of how to use them, built up over years of experience (Schön, 1983). This process is popularly characterized by the UK Design Council's *double diamond* (Sharp et al., 2023), highlighted in Figure 2. Initially when a designer receives a specification, they begin to explore *divergently* how to think about the problem: this involves reasoning about the materials, context, people, social structures, and policy context of the request (Tomitsch et al., 2020). Subsequently, they begin to *converge* on a more concrete definition of the problem and present it to the stakeholders involved. At this moment, all stakeholders *diverge* again, exploring various designs without limits as they explore the potential solution space. Finally, the designer converges on a solution, synthesizing all that they have learned to present a design that is on time, budget, and to specification. The double diamond merges a designer's expertise with their situated knowledge and

experience.

The outcome-centered perspective inherent in the double diamond brings with it the notion that a design should be finished and then deployed in its "finished state" (Tonkinwise, 2004; Redström, 2017). Here we find an interesting bridge to current embodied artificial agents: they too pass through a phase of training and are only subsequently deployed when they have reached a pre-defined threshold of performance. In interaction design, this perspective limits a finished design to its intended function. Despite the efforts of human factors, user-centered design and participatory design methods (Sharp et al., 2023), ethnographic studies often reveal the user to be *constantly* spending time and creative energy to configure these finished designs and their intended functions into their own lives (Dourish, 2001; Suchman, 2006; Dörrenbächer et al., 2022; Norman, 2010). This has lead to the increasingly blurred line between what constitutes a designer and a user of technology (Redström, 2017).

**Research through design (or the continuous double diamond)** The field of human-computer interaction (HCI) has seen in the last two decades the rise of *research through design* (RtD) (Koskinen, 2011) which supports the notion that design is never finished. It is commonly framed as "an active process of ideating, iterating, and critiquing potential solutions, design researchers continually reframe the problem as they attempt to make the right thing" (Zimmerman et al., 2007). At its core, RtD can be thought as a continuous double diamond (see Figure 2), with its tail (problem) and head (solution) lopped off. The design process then becomes reflective: where the morals, lived experience, and aesthetic preferences of the designer[7] can inform their professional training (La Delfa et al., 2020), leading to completely new (divergent) ways of interacting with technology (Bewley & Boer, 2018), or familiar (convergent) twists on existing ones (Odom et al., 2019). We would like to emphasize that a person does not require formal design training to practice RtD: there are numerous examples from RtD that invite the user to instantiate co-existing relationships with technology, examples of which can be found in Appendix C.

### 4.3. From Elsewhere to Embodied Agents

By exploring co-existence in biology, we have shown that living organisms leverage the physical world to offload the need for encoding all necessary information for survival and action, while also enabling diverse and adaptable behaviors. By exploring co-existence in design, we have highlighted RtD as a promising approach to balance convergence and di-

---

[7]As a *first person method* (Loke & Schiphorst, 2018) RtD can trace its theoretical foundations back to embodiment (Lakoff & Johnson, 1985).

vergence in the interaction between humans and technology. These ideas can be naturally extended to embodied agents: leveraging the situated knowledge in the environment and in the human user enables embodied agents to successfully change, evolve and interact in a meaningful way within their specific niches.

## 5. Alternative Views to Co-existence

**AGI/ASI vs. co-existence**   While co-existence is a goal and property in itself, other positions argue for different goals and capabilities of long-term interactive artificial agents within our societies. Paolo et al. argues in favor of attempting to achieve artificial *general* intelligence (AGI), describing the goal as "creat[ing] intelligence that either parallels or exceeds human abilities". For this goal, they state that embodiment and situated intelligence are essential conditions for achieving AGI. Similarly, Hughes et al., argue in favor of artificial *superhuman* intelligence (ASI) and propose open-endedness as a prerequisite to ASI. Whilst we share an understanding of the importance of embodiment and open-endedness, neither position requires mutual co-shaping for the widespread application of artificial agents in human society. Despite its risks (Naudé & Dimitri, 2020; McLean et al., 2023), proponents of AGI and ASI point to the accelerated progress and benefit for humanity driven by a single superior intelligence. Instead, we believe that through the increase in diversity, co-existence aims for something more beneficial and robust: we place meaningful and reciprocal interactions with humans at the center of our proposal.

**Unilateral alignment vs. co-existence**   Yang et al. state that "unified alignment between agents, humans and their environment" is key to the success of agents in real-world applications. They propose that agents not only align with human users, but also with the environment and the agent's own constraints. Furthermore, they highlight the difficulty of discovering human intentions due to partial observability, temporality and stochasticity. Although they discuss the need for agents that can align with evolving preferences, a process they denote as continual alignment, they still assume that preferences are something that is known by the human a priori. They write: "the tasks assigned by humans can be viewed as the initial inputs to the working system (especially to the agents), which reflects the underlying goals and human intentions". We instead believe that the human's goals are formed through interacting with the agent.

## 6. Towards Co-existing Embodied Agents

We have seem how both human and non-human organisms evolve and coexist within their own niches. What can the machine learning community learn from these processes?

This section outlines key research directions toward developing co-existing embodied artificial agents, focusing on three fundamental aspects: the principles responsible for shaping co-existence (what), the hardware that supports it (where), and the methods that may enable it (how). Finally, we address some ethical considerations of co-existence.

### 6.1. What fosters co-existence?

Achieving co-existing embodied agents goes beyond engineering and optimization; it requires principles that shape their evolution, integration, and interaction with users. This section highlights two key principles: open-endedness for continuous adaptation, and the user as a designer, whose situated knowledge facilitates the development of the agent.

**Open-Endedness**   Hughes et al. (2024) argues in favor of open-endedness to design continuously evolving agents, defining it as a property of systems that produce *novel* and *learnable* artifacts from the perspective of an observer. We agree that open-endedness is essential to achieve co-existing agents, and highlight the shared importance of the observers perspective between open-endedness and RtD. We see the role of the observer as a driver of continuous change and exploration, not just a creative optimizer for a specific task.

**User as the Designer**   Often the user is seen as someone who should not have to deal with the complexities that arise from interacting with technology (Norman, 2010). In RtD, this perspective is rejected in favor of seeing the user as someone who has situated knowledge, or is a *connoisseur* of their situation (Zimmerman et al., 2007; Loke & Schiphorst, 2018). This knowledge, including tacit, institutional, craft or social knowledge, can help them mediate the agent's situatedness. We argue that this perspective is essential to co-existence and should guide agents development.

### 6.2. Where do we foster co-existence?

We envision the engagement of the user as a co-shaper of an agents sensing and acting capabilities, both around the agent and within the agent itself.

**The space around the agent**   Consider an instance of an agent using an *inside-out* navigation system, such as SLAM (Durrant-Whyte & Bailey, 2006) which is inherently prone to drift. If an *outside-in* navigation system (such as a Mocap system) is instead used, the agent can be designed such that the situated human can configure the placement of the beacons. Whilst this sounds like a poorly designed system that requires constant maintenance[8] research on AI education has favored this practiced-based approach, as it

---

[8]As a comparison, we would like to highlight the resources required to create and curate large-scale datasets.

fosters a kind of tacit understanding of the capabilities and limitations of the system (Flechtner & Stankowski, 2023). This situated knowledge from the human user can help an embodied agent co-exist in its environment.

**The morphology of the agent** Evolutionary robotics has demonstrated that by changing the morphology of an artificial agent, you change their capabilities and limitations (Pfeifer & Bongard, 2006). Additionally, advancements in manufacturing technology is rapidly expanding the potential forms an agent could take (Kriegman, 2020). This concept has been explored in the context of human-drone interaction. La Delfa et al. gave users a drone that could initially only hover in place. By moving with the drone, the users were able to selectively expand its perceptive field. As the field grew in size, unique patterns of interaction emerged based on its the shape and size. The mutability of the drone's sensory field allowed for a meaningful relationship to evolve.

### 6.3. How can we foster co-existence?

In their current form, even approaches designed to overcome the assumption of a static optimization problem (e.g., as reinforcement learning, meta-learning, and continual learning) seem unable to foster co-existence[9]. Instead, we urge the community to explore two key directions: (i) leveraging foundation models as external sources of knowledge rather than end-to-end solutions, and (ii) integrating human-in-the-loop learning with evolutionary algorithms.

**Embracing foundation models as external** Recent methods have used foundation models or composite systems that incorporate foundation models to generate agent behavior (Brohan et al., 2023). While using these models directly as policies is not sufficient for co-existing agents, foundation models still have valuable properties that can be leveraged (even if these are currently prone to hallucinations (Li et al., 2023b; Zhang et al., 2023)): they can act as an external storage of *generic* knowledge that an agent could query for bootstrapping purposes *without replacing situated knowledge*. This external knowledge could help decrease the memory and computation requirements to build embodied agents. Additionally, foundation models could serve as external teachers to agents to bootstrap their performance (Yang et al., 2024a) and guide exploration (Kumar et al., 2024) *without replacing situated exploration*. While we understand these models can also be used for multimodal perception and reasoning, we highlight the risk of embed-

ding such internal components of embodied agents with generic and stagnant knowledge and encourage researchers to consider using the real world "as its own best model".

**Learning and evolving with humans as we go** To enable mutability and speciation we advocate for *human-in-the-loop* learning with *evolutionary* algorithms. Evolutionary algorithms (Bäck & Schwefel, 1993; Li et al., 2023a) can maintain diverse candidate solutions throughout the (continuous) learning process, allowing agents to execute multimodal behavior, both divergent and convergent (Mouret & Clune, 2015). When combined with interactive learning paradigms (Zanzotto, 2019; Mosqueira-Rey et al., 2023), such as by using preferences or demonstrations, the evolution process can also be progressively shaped through meaningful interactions with the human, allowing the agent to deal with evolving goals and expectations.

### 6.4. Should we foster co-existence?

It's important to state that co-existence gives users the ability to shape and be shaped by embodied agents. This carries the inherent risk of manipulation of the agent's behavior by malicious users and vice versa. We highlight the importance of developing agents that have the ability to recognize harmful behavior and respond in a manner that upholds safety, fairness, and accountability. By giving users the responsibility to shape the agents in their environment enables them to do so in their own particular way. This results in a heterogeneous population of bespoke agents. Local and diverse groups have been shown to exhibit strong innovative capabilities and pro-social behavior, both in human and AI collectives (Lai et al., 2024). We anticipate that groups including co-existing agents will have similar properties, leading to an increase of the quality of their systems.

## 7. Conclusion

In this paper, we have argued that the current paradigm for designing embodied artificial agents is fundamentally ill-suited for meaningful, long-term human interaction. We proposed *co-existence* as a new paradigm for the design of embodied agents that emphasizes meaningful, reciprocal interactions sustained over time. Drawing from biology and design, we showed how human and non human organisms leverage the physical world in convergent and divergent ways. We outlined key research directions for co-existing agents, emphasizing open-ended, human-in-the-loop learning and the user's role in shaping both behavior and morphology. We envision a future where artificial agents don't just exist but *co-exist*, actively shaping and adapting to humans and their environments.

---

[9]Reinforcement learning assumes a fixed reward structure in the learning problem; meta-learning adapts within a predefined (fixed) meta-distribution of possible scenarios the agent might encounter; continual learning instead mitigates catastrophic forgetting, yet does not reason about *unknown unknowns* (Lehman et al., 2025). For an extended argument on why currently these methods fail in the real-world, we refer the reader to Lehman et al. (2025)

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

## A. Additional Notes on the Definition of Co-existence.

**Closed system**   For simplification, we have implicitly assumed that our system is *closed*, meaning that the quality of the interaction is only influenced by the elements within the system (environment, human and agent). We have also assumed that there is a single human user in the system. However, we can easily extend this to open systems and multiple users by considering the correspondent interaction terms with additional elements external to the system (e.g., external societal rules, other members of a sport team), without a change on the definition of co-existence. We expect that the effect of these additional interactions to be dependent on each specific system.

**Evolution of the quality of a system**   We would also like to highlight that we do not expect the quality of the system to be monotonically increasing over time – in fact, *we argue that it should not*. As discussed in Section 3.2, co-existing agent must have the ability to explore and exploit divergent behaviors that may only increase the quality of the system *in the long term*.

**Nature of $Q_S$ and $T_S$**   Finally, like all the elements in the system, the operationalization and interpretation of the quality function $Q_S$ is dynamic (meaning it changes over time) and specific to every system. The same can be applied to the time horizon $T_S$: each particular system should have, even if implicitly, a specific time horizon to access the evolution of the system itself.

## B. Are Current Embodied Agents Already Co-existing?

Naturally, one might question whether current embodied agents are already *co-existing* with humans. In this section, we present examples and discussions on key challenges preventing current agents from being co-existing.

**Social Robots**   A prominent example of embodied agents designed for human interaction are *social robots* (Breazeal et al., 2016; Leite et al., 2013). Companies like Jibo and Anki introduced social robots to the market with high expectations, only to face eventual failure (Tulli et al., 2019). A significant factor contributing to this is the challenge of sustaining long-term interactions by current embodied agents. Without the ability to change through interaction and become situated into their environment, social robots remain ill-suited for prolonged use. They often succumb to the *novelty effect*, where user engagement diminishes over time as the robot's initial appeal wears off (Reimann et al., 2023).

**Bias-amplifying interaction**   Large language models have been widely integrated into the architecture of embodied agents (Xiang et al., 2024; Driess et al., 2023). These models have now been widely adopted by diverse user groups. While most AI systems influence human behavior, they themselves do not retain user-driven modifications beyond the immediate context window. This lack of adaptability is already problematic, as user-provided knowledge is not incorporated. Worse, studies have shown that interacting with slightly biased AI systems can amplify biases in users, an effect not observed in human-human interactions (Glickman & Sharot, 2024). These systems not only fail to adapt through interaction, reinforcing a unilateral dynamic, but they also degrade overall system quality by increasing bias in users. As LLMs are increasingly integrated into interactive robots, these issues are likely to persist, if not worsen, through prolonged human-robot interactions.

**Just turn on the light**   Consider a robot designed to tidy up homes and offices by identifying, classifying, and sorting objects. In industrial settings, similar robotic failures require expert technicians to debug classifiers, diagnose issues, and retrain models with additional data, such as images captured under varied lighting conditions. However, relying on expert interventions is impractical for home-deployed robots. A more viable solution is for robots to make use of humans situated knowledge within their environment. Humans understand their space and might recognize how the specific lighting affects object classification. Instead of requiring an expert to retrain the system, a robot could ask for help, prompting users to turn on the light and even learning that turning on lights improves classification performance. By adapting through situated interactions, the robot avoids repeated failures and reduces the need for costly expert intervention and large-scale data collection.

## C. Potential Co-existing Technology Today

In this section, we highlight several examples of technology with properties that foster co-existence.

**Blo-Nut: A Mutable Interaction Interface**   Figure 3 shows Bewley & Boer's "Blo-Nut", a silicone doughnut that affords the user a blank slate to interact with. The object inflates and deflates and can be programmed to music. It's non-humanoid shape affords interactions to the human in ambiguous ways, which Sandry argues, is an opportunity to build effective communication between humans and artificial agents.

**Motorid: a Shape-changing and autonomously balancing motorcycle**   Figure 4 shows Yamaha's "Motorid", a shape changing, self balancing motorcycle (Hara et al., 2021). It has a twisting chassis and autonomous driving

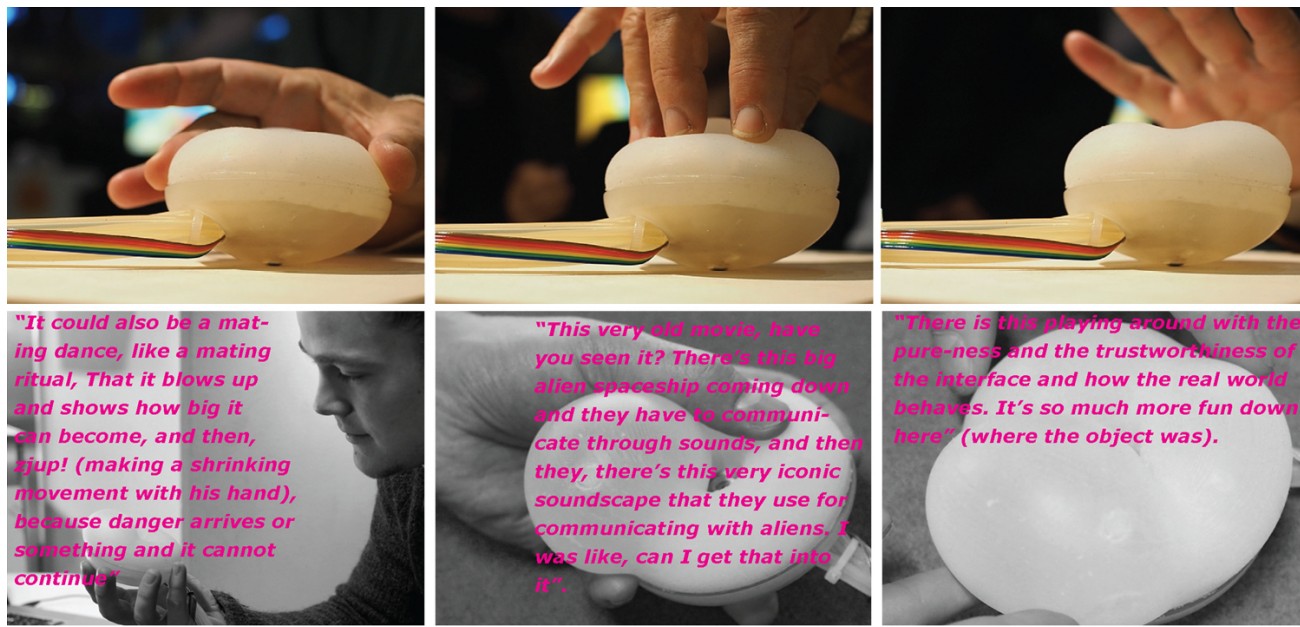

*Figure 3.* "Blo-Nut" is a silicone doughnut that affords the user a blank slate to interact with (Bewley & Boer, 2018). The object inflates and deflates and can be programmed to music.

abilities that influence how riding the motorbike feels in real time. This dramatically changes motorcycling from its culture to its engineering principles. Whilst not a child of the RtD method, but rather a concept bike, it balances divergent and convergent themes. Blurring the definition of what is a bike and an autonomous agent.

**Mutable Perceptive Fields in Human-Drone Interaction**
Figure 5 shows how humans can shape the perception of embodied agents. Their size and shape played a role in shaping how the users flew the drones and how they made meaning with them (La Delfa et al., 2024b).

**Mutable Morphology and Locomotion**    Figure 6 shows how morphology can be changed and recover from damage, re-learning how to walk Kriegman et al. (2019). The agent learns how to walk through periodically inflating and deflating its individual cells, exploiting it's own physical shape. Although this does not involve a human user, it demonstrates the value of mutable morphologies. For example, we see great potential in mutable morphology to express various mannerism through different gaits. Especially in the context of Vergunst & Ingold's work on the contextual nature of walking. Thus culminating in rich, heterogeneous populations of artificial agents at scale.

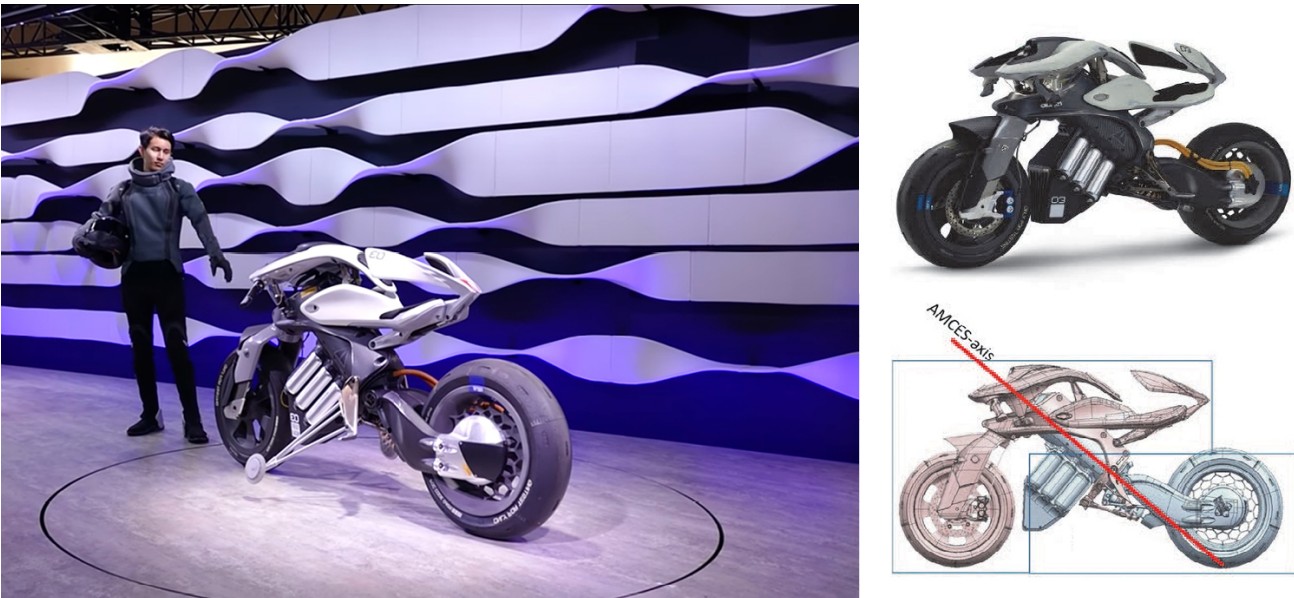

*Figure 4.* Yamaha's "Motorid" is a shape changing, self balancing motorcycle (Hara et al., 2021). Its unique twisting chassis is able to affect the ride feel in real time as well as drive autonomously.

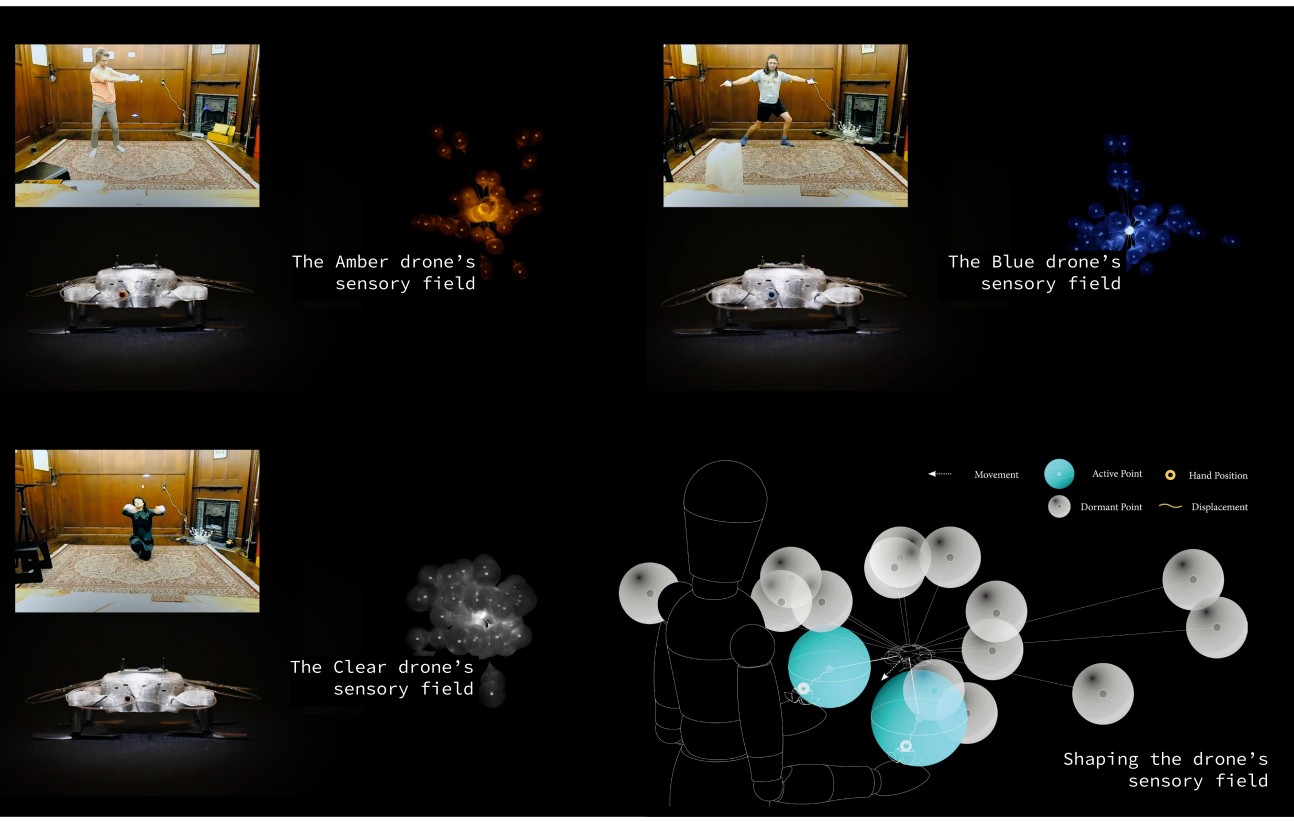

*Figure 5.* "How to Train Your Drone" (La Delfa et al., 2024b): depicted here in orange, clear and blue are the sensory fields of the drones. By interacting with the drone, its sensory field can be changed with human intention. However the consequences of such changes are not always predictable. This work demonstrates the potential of interacting with the the sensing and acting capabilities of mutable agents.

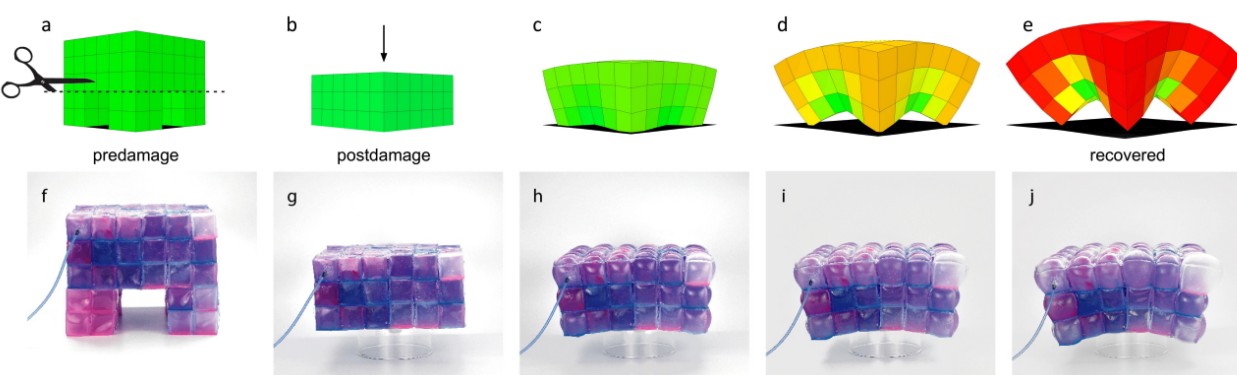

*Figure 6.* Self recovering locomoting voxels (Kriegman et al., 2019): by virtue of an evolutionary algorithm, the agent is relearning how to walk by changing the inflation patterns of its individual cells.

