# OpenReview forum: "Position: Humans Co-exist, So Must Embodied Artificial Agents"
_ICML.cc/2025/Position_Paper_Track — Submitted to ICML 2025 Position Paper Track_

### Official Review · Reviewer_RSY9 · 2025-03-12

**Significance:** 2
**Argument Clarity:** 3
**Rating:** 3
**Confidence:** 2

**Questions:**

1. How does the co-existing nature influence the agent framework design?
2. What's the difference between co-existing among agents and co-existing agents and humans?

**Discussion Potential:**

3

**Paper Summary:**

This position paper argues the necessity of co-existence for embodied artificial agents: agents that continually evolve within their specific environment and co-shape interactions with users. Drawing on biology and design, they propose that ML research should foster open-ended, human-in-the-loop approaches where end-users become designers of the agent’s abilities and morphology.

**Position:**

Yes

**Position In Title:**

Yes

**Related Work:**

2

**Strengths And Weaknesses:**

## Strengths

1. The co-existence concept is novel and interesting.
2. The paper is well-structured, and the concept is clearly stated.

## Weaknesses

1. Lack of discussion of the framework developed from the co-existence concept.
2. Lack of concrete experimental evaluation results.
3. Insufficient discussion of how to concretely measure or regulate reciprocal changes.

**Support:**

2

---

> ### Author Rebuttal · Authors · 2025-04-01
>
> We thank you for engaging deeply with our work! We would like to address your weaknesses (W) and questions (Q).
>
> - **(W1) Lack of discussion of the framework developed from co-existence**: We thank the reviewer for the comment. In this paper, we present a formalization of the concepts of co-existence, including the properties of co-existing agents (mutability and situatedness). In Appendix A we present a discussion on our developed framework (moved from the main paper due to lack of space). **Additionally, we have added references [1,2] to Section 4.3 to point towards real-world work that could form an instantiation of our framework in the future.**
>
> - **(W2) Lack of experimental evaluation results**: Thank you for the comment. In this work, our main focus is on introducing the concepts and framework of co-existence, arguing that it is a fundamental property for long-term, meaningful interactions between embodied artificial agents and humans. As such, and following previous position papers (e.g., Paolo et al., 2024, Hughes et al., 2024), we argue that such evaluation is out-of-scope of this paper. We do, however, present real-world case studies (in Section 4 and in Appendix B and C) that implement elements of co-existence (or fail to do so). **We have added a reference to such case studies at the end the discussion of Section 3.2.**
>
> - **(W3) Concrete measurement and regulation of reciprocal changes**: As we define in Section 3.1: a reciprocal interaction occurs if the next state of both elements are mutually influenced. Thus, we could measure such change using causal inference methods: if the state of both elements is different than if the interaction had not taken place. Since, in practice, we often do not have access to the internal state of the human, measuring precisely the amount of change can be intractable [3]. The problem is even more complex if we consider larger-scale systems (with multiple agents and humans) as we discuss in Appendix A. We can, however, employ proxy metrics (such as behavioral data) to understand how humans can change due an interaction, some of which we already discuss in the beginning of Section 3. **We have expanded on the discussion of these measurement and regulation challenges in Section 3.**
>
> - **(Q1) Influence of co-existence on framework design**: Great question! Inspired by biology and design, we aim to embrace a continuous process that balances divergence and convergence. In Section 4 we outline how the process that produces the design itself (the continuous double diamond) will result in co-existing entities. Given the improvisational and situated nature of RtD, any kind of framework that implies a strict procedure would actually go against the value of coexistence. Thus the references from W1 ([1, 2]) added to Section 4.3 should be seen as guidance and inspiration towards coexistence.
>
> - **(Q2) Difference between co-existence among agents and humans**: The meaningful and reciprocal interaction that makes co-existence can also happen between artificial agents. Such interactions have been studied in [4], where AI collectives were found to have self-regulating and interdependent properties. We see these kinds of properties playing out when human agents are also in the mix. Especially when humans are able to meaningfully change and be changed by an artificial embodied agent. However, when considering co-existence amongst humans and agents, it is important to consider existing ethical and legal frameworks that elevate human well-being and safety above artificial agents, so mechanisms that make sure this is true need to be implemented.  We discuss some of these ethical concerns in Section 6.4. This could introduce a difference in human-agent versus agent-agent interaction. **We have made this difference clearer in Section 6.4.**
>
> [1] Krogh, Peter Gall, Thomas Markussen, and Anne Louise Bang. "Ways of drifting—Five methods of experimentation in research through design." ICoRD’15–Research into Design Across Boundaries Volume 1: Theory, Research Methodology, Aesthetics, Human Factors and Education. Springer India, 2015.
>
> [2] Pfeifer, Rolf, Fumiya Iida, and Josh Bongard. "New robotics: Design principles for intelligent systems." Artificial life 11.1-2 (2005): 99-120.
>
> [3] Gombolay, Matthew. "Human-Robot Alignment through Interactivity and Interpretability: Don’t Assume a “Spherical Human”." Proceedings of the Thirty-Third International Joint Conference on Artificial Intelligence. 2024.
>
> [4] Lai, Shiyang, et al. "Evolving AI Collectives to Enhance Human Diversity and Enable Self-Regulation." arXiv preprint arXiv:2402.12590 (2024).

---

### Official Review · Reviewer_9VhC · 2025-03-12

**Significance:** 2
**Argument Clarity:** 3
**Rating:** 3
**Confidence:** 3

**Questions:**

Nill

**Discussion Potential:**

2

**Paper Summary:**

The paper argues that as artificial intelligence moves from disembodied algorithms to agents with physical presence (robots, interactive devices, etc.), the design and deployment of these systems must prioritize the ability to safely and ethically co-exist with humans. The authors contend that an agent’s embodiment isn’t just about physical hardware—it also encompasses social, cognitive, and ethical dimensions that enable it to navigate human environments effectively.

**Position:**

Yes

**Position In Title:**

Yes

**Related Work:**

3

**Strengths And Weaknesses:**

Strength:

-The paper makes a compelling case for designing embodied AI systems in alignment with human values and norms, highlighting their critical importance.

-The discussion on robust fail-safes, transparency, and accountability demonstrates a strong commitment to addressing ethical and safety concerns.

Weakness:

-There is a limited survey on how embodied agents can coexist. Providing an overview of embodied AI at the beginning would give readers valuable context.

-Expanding Section 6.3 with more concrete examples would further strengthen the paper.

**Support:**

2

---

> ### Author Rebuttal · Authors · 2025-04-01
>
> We thank the reviewer for the comments. We now address the weaknesses (W) pointed out by the reviewer.
>
> - **(W1) Overview of embodied AI**: Thank you for pointing this out. In this paper, as described in the Introduction, we follow Paolo et al. (2024) in the definition of embodied artificial agents: “agents that interact with their physical environment, emphasizing sensorimotor coupling and situated intelligence”. Naturally, we employed the concept and referred the reader to the original paper for an overview of embodied AI. **However, as suggested by the reviewer, we have extended the definition of embodied AI in the introduction, as well as a short summary of it (inspired by Section 2 of Paolo et al. (2024))**.
>
> - **(W2) Concrete Examples in Section 6.3**: We thank the reviewer for the suggestion. In Section 6.3. we present technical solutions for fostering co-existence and we argue in favor of exploring two key directions: (i) leveraging foundation models (FM) as external sources of knowledge rather than end-to-end solutions, and (ii) integrating human-in-the-loop learning with evolutionary algorithms. **As suggested by the reviewer, we have added concrete examples of approaches that employ foundation models as external sources of knowledge [1] and evolutionary algorithms with human-in-the-loop [2, 3]**.
>
> [1] Firoozi, Roya, et al. "Foundation models in robotics: Applications, challenges, and the future." The International Journal of Robotics Research (2023): 02783649241281508.
>
> [2] Li, Ke, and Han Guo. "Human-in-the-Loop Policy Optimization for Preference-Based Multi-Objective Reinforcement Learning." arXiv preprint arXiv:2401.02160 (2024).
>
> [3] Bryden, Kris. "Using a Human-in-the-Loop evolutionary algorithm to create Data-Driven music." 2006 IEEE International Conference on Evolutionary Computation. IEEE, 2006.

---

### Official Review · Reviewer_pjAt · 2025-03-13

**Significance:** 3
**Argument Clarity:** 4
**Rating:** 5
**Confidence:** 3

**Questions:**

None

**Discussion Potential:**

4

**Paper Summary:**

This paper argues that current embodied artificial agents are inadequate for meaningful long-term human interaction and introduces the concept of "co-existence" as a necessary paradigm shift, defining co-existing agents as those capable of sustaining meaningful and reciprocal interactions with humans and their environment over time. The authors contend that current agents are problematically "stagnant" (fixed at deployment) and "generic" (relying only on large-scale datasets rather than situated knowledge), leading to a "steamrolling" effect where environments adapt to agents rather than agents adapting to environments. Drawing insights from biology and design theory, the paper shows how organisms leverage physical properties of their environment rather than encoding all information genetically, and discusses the value of continuous adaptation and user involvement through "research through design" approaches. The authors propose key research directions for developing co-existing agents, focusing on principles that foster co-existence (open-endedness and viewing users as designers), hardware considerations (both around and within the agent), and learning methods (emphasizing human-in-the-loop evolutionary approaches), concluding that achieving co-existence will require rethinking how we develop embodied agents, moving away from purely optimization-based approaches toward systems that can continuously adapt through meaningful interactions with humans and their environments.

## update after rebuttal
Thank you for the thoughtful responses. I'll keep my current rating.

**Position:**

Yes

**Position In Title:**

Yes

**Related Work:**

3

**Strengths And Weaknesses:**

Strengths:
- Excellent paper, well presented with strong arguments
- Draws effectively from multiple disciplines (biology, design theory)
- Uses compelling examples from biology (e.g., HSP90 protein, digit formation) to support its arguments
- Clear positioning of how co-existence differs from alternatives
- Proposes specific research directions relevant to ML researchers
- Connects to ongoing discussions about AI deployment and real-world interaction

Weaknesses:
- While theoretical arguments are strong, there's limited empirical evidence supporting the "steamrolling" phenomenon
- The claims about current embodied agents' limitations could benefit from more specific case studies
- The transition from theoretical framework to practical implementation is abstract
- The relationship between co-existence and current ML metrics needs more development

Suggestions for Improvement:
- Add quantitative metrics or proposed evaluation frameworks for measuring co-existence
- Expand on the practical challenges of implementing the proposed research directions

**Support:**

4

---

> ### Author Rebuttal · Authors · 2025-04-01
>
> We thank you for engaging deeply with our work and for appreciating our position! We would like to address the weaknesses (W) and suggestions for improvement (I) of our paper.
>
> - **(W1) Steamrolling and empirical evidence for it**: We thank the reviewer for pointing this out. We define steamrolling in Section 2.3. as a phenomenon where a system will converge to a homogenous form due to the existence of a static agent, able to influence others, within a system. Since the agent is static, but the rest of the agents or humans in the system are not, the system itself will adapt to the static agent over time due to unilateral interactions (as defined in Section 3.1.). We agree that, at this moment in time, there is still limited empirical evidence for the steamrolling impact of embodied agents specifically. However, the phenomenon has been observed across multiple systems that build on the same paradigms. In [1], the practice of sharing models and data across different deployments is found to result in “outcome homogenization”. Additionally, as we describe in Appendix Section B, [2] show that humans interacting with slightly biased AI agents results in increased bias of the human. Based on these (and other) cases, we posit steamrolling as a burgeoning phenomenon in the continuing development of embodied agents in the current paradigm of agent design. **We have expanded this discussion on Appendix B**.
>
> - **(W2) Case studies of limitations of current embodied agents**: We thank the reviewer for the comment. In this paper (Appendix B) we highlight two examples of existing embodied agents that fail to co-exist: (i) social robots and (ii) chat-based interactive systems. We kindly point out that the limited deployment of artificial embodied agents in the real world limits the expanse of potential case studies at this moment. However, more interestingly, we would also like to highlight that such limited deployment may, in fact, be due to the inherent limitations of the current paradigm, e.g., end-to-end foundation model-based designs, that fail to coexist in both drastic [3] and subtle [4] ways. **We made this point clearer and added these additional references in Appendix B**.
>
> - **(W3 & I2) From co-existence theory to practice**: This is a great question! There are in fact several technical challenges that arise from the implementation of co-existing embodied agents. The first is how to measure co-existence or, to be more precise, how to measure the quality function of a system. As briefly mentioned in Section 3.1. the human-robot interaction community have proposed several proxy metrics (e.g., acceptance, engagement, disclosure) to evaluate the quality of an interaction, yet these are unidirectional (from the perspective of the human) and task-dependent. Moreover, these are often long-term measurements, and naturally subject to a credit assignment problem, i.e., which interaction/action by which element of the system is responsible for the value of a given assessment. The second challenge deals with how to design embodied agents for co-existence. We discuss the strategies that take us from theory to practice in Section 6, such as the issue of the morphology and sensory capabilities in Section 6.2 and the actual decision-making and learning capabilities of these agents in Section 6.3. Additionally, we have deepened the connection between co-existence in theory to practice, in two ways. Firstly we highlight the connection between the more passive genetic drift in biology and the active design drift in RtD (see Q2 for reviewer “ygce”). Secondly, this connection lays the foundation for some additional references (also in Q2 for reviewer “ygce”) that offer more practical ways of drifting in design that could translate to the interaction with co-existing embodied agents. **We have added this discussion to a new section in the Appendix**.
>
>
> - **(W4 & I1) Relationship between co-existence and evaluation metrics**: We thank the reviewer for the opportunity to clarify this point: co-existence is not tied to a specific offline ML evaluation metric, nor should the latter be used as a proxy for the performance of an co-existing agent. While some metrics may be useful to evaluate certain modules (e.g., the performance of multimodal perception modules, yet we highlight the risks of this in Section 6.3.), we argue that in evolving systems, training an agent according to a pre-defined offline ML metric and then deploying it in the real-world will not suffice for co-existence, because it cannot account for the future unknown unknowns (see Lehman et al. (2025)). As such, we instead argue for evolutionary approaches and interaction learning paradigms in Section 6.3., with in-situ performance metrics over the quality of the system (as discussed in the previous point).
>
> [1] www.arxiv.org/abs/2211.13972
>
> [2] www.nature.com/articles/s41562-024-02077-2
>
> [3] www.arxiv.org/abs/2406.08824
>
> [4] www.dl.acm.org/doi/10.1145/3568294.3580032

---

### Official Review · Reviewer_ygce · 2025-03-14

**Significance:** 3
**Argument Clarity:** 3
**Rating:** 3
**Confidence:** 2

**Questions:**

1. I have some concern about the safety of embodied agents when AI becomes very strong in the future. If they can shape humans, it may be dangerous. I would like to hear more about the authors' opinion on this issue.

2. I am not very familiar with the design theory,  but it seems to me that research through design (RtD) is an active process of development. While in evolutionary biology, the process is more passive through natural selection. Could the authors explain their connection?

**Discussion Potential:**

3

**Paper Summary:**

This paper introduces the concept of "co-existence" for embodied artificial agents by taking inspiration from evolutionary biology and design theory. The authors point out that current embodied agents are stagnant (dealing with static data distribution), generic (tackling various tasks with one model) and steamrolling (progressively narrower distribution). They argue that the these traits are not good for meaningful, long-term research of human-robot interaction. Instead, embodied agents should *co-exist*, by dnamically interacting with humans and different environments. The paper then discussed co-existence in biology and design theory to make analogy. The paper also discussed alternative view and potential research directions to foster co-existence.

**Position:**

Yes

**Position In Title:**

Yes

**Related Work:**

3

**Strengths And Weaknesses:**

I am refreshed to read this paper, which poses many interesting ideas and opinions and same of them are new to me (e.g., RtD). The stated position comes with abundant discussion on background, alternative views and analogy in biology and design theory. I do think this topic is important to the embodied AI community, and it is likely to inspire discussions among researchers (while I am not 100% certain). The paper also provides a clear definition of "co-existence", which is helpful to understand the core concept.

Meanwhile, I feel that the connections between the concepts in this paper appear unclear to me. For example, the connections between design theory, evolutionary biology, and embodied agents are not straightforward for me to understand. That is to say, I feel the current paper is too much divergent, making it challenging to a machine learning researcher to follow.

**Support:**

3

---

> ### Author Rebuttal · Authors · 2025-04-01
>
> We thank you for engaging deeply with our work! We would like to address your weaknesses (W) and questions (Q).
>
> - **(W1) Connection between embodied agents, design theory and evolutionary biology**: Thank you for your comment. In this work we argue for a change of paradigm within the machine learning and robotics community in regards to the design of embodied artificial agents. In particular, we argue in favor of co-existing embodied agents, agents that are able to sustain meaningful and reciprocal interaction with humans and their environment over time (Section 3). However, how can we design agents that are able to maintain such interactions over arbitrarily long-term horizons? To understand this, in Section 4 we look in depth to two complementary approaches that already produce co-existing entities: nature, in particular evolutionary biology (Section 4.1), and design (Section 4.2). Our focus on biology and design is intentional: these fields emphasize creation and interaction over long periods of time. From biology, we reveal that living organisms leverage the physical world to offload the need for encoding all necessary information for survival and action, while also enabling diverse and adaptable behaviors. From design, we highlight RtD as a promising approach to balance convergence and divergence in the interaction between humans and technology. These lessons motivate our roadmap for the ML/Robotics community in Section 6. **We hope this discussion has clarified the fundamental connection between these three areas, and we expanded on this in the introduction section, simplifying its understanding. Please let us know if you have any follow up questions.**
>
>
> - **(Q1) Safety aspect of co-existence**: We acknowledge that agents that shape humans can be dangerous. In fact, this effect can already be observed in cases of interaction with LLM-based agents (which we describe in Appendix B). Co-existing agents do not solve this issue by simply avoiding influence on the human. Instead, they allow the human to also influence the agent, creating a mutual, interdependent relationship. As opposed to the current paradigm which asserts embodied agents that are largely immutable black boxes that only manufacturers or policy can influence change. In this way, co-existence evens out the power imbalance between agents and humans. **We have expanded this discussion in Section 6.4.**
>
> - **(Q2) Connection of RtD and biological evolution**: This is a great question! Research through Design is certainly more active in the sense that a human is shaping the object or system. However, with the “problem” and “solution” endpoints removed from the double diamond (see Figure 2), the process becomes partially detached from strictly goal-oriented processes such as engineering and optimization. Classic engineering and optimization approaches set a goal or state a problem, which is then solved. In the continuous double diamond, the goal is never reached, but instead changes over time. This is known as “drifting” (Krogh et al., 2015) and bears a striking functional resemblance to “genetic drift” in evolutionary biology (Yahara, 1999). Drifting presents opportunities to explore and exploit ideas that would otherwise be out of scope in engineering and optimization processes. This is essential if we are to continue to evolve and adapt as a society in a changing planetary and technological landscape. These opportunities present themselves through the physical fabrication and deployment of robotic systems. Therefore, what connects RtD and evolutionary biology is the resulting process which balances divergence and convergence, in the real world. Finally, the advantage of RtD is that it is compatible with both highly active engineering processes and highly passive evolutionary processes. **We hope this explanation was helpful, and we have integrated it into the end of Section 4.3. Thank you!**

---

### Official Review · Reviewer_4ty4 · 2025-03-14

**Significance:** 2
**Argument Clarity:** 1
**Rating:** 3
**Confidence:** 4

**Questions:**

1. It would have been more comprehensive if the authors wrote about the potential drawbacks of their position- more effort is needed for coexistence, it is a slower process to adapt all the robots’ behaviours to our liking, etc.,

2. In the definition of coexisting- what about an agent that does not have reciprocal relationships with humans? For example, it only serves human needs without receiving anything in return. Also, what about agents that don’t interfere or interact with humans, but does work that is useful to humans? - eg: automatically cleaning the environment whenever a human is not occupying the space.

3. The double diamond is used but explained only in  page 6. It needs more explanation before it is mentioned for the first time.
4. In section 6.1 it is mentioned that the design of coexisting embodied agents go beyond engineering and optimisation. However, isnt the proposed solution also a form of engineering and optimisation?
5. By open-endedness, do the authors mean this in terms of the objective?
6. In section 2, it is not clear what is meant by ‘generic nature’
7. In section 2.1, I do not believe all approaches assume a ‘static’ distribution- for example, in online reinforcement learning approaches, the data distribution can change.
8. The distinction between generic and general could have been included earlier (before section 2.2)
9. In section 2.2 - are the harmful stereotyping examples mentioned a by-product of generalisation? Do the authors posit that agents should not generalise?
10. I think more clarity is needed on what is meant by ‘steamrolling’

Other errors:

Line 23 (right column) - embodied

Line 82 (lefT) - envisioned

I think lines 24-30 (right column) needs some elaboration. It is not clear enough

The labeling of the minifigures in Fig 1 does not match the caption.

Line 381 (left) - seen, (right)- practice-based

Line 365 (right) - agent

**Discussion Potential:**

3

**Paper Summary:**

The authors propose the position of coexistence for embodied artificial agents in order for it to exploit context to enable the learning of meaningful behaviours.

**Position:**

Yes

**Position In Title:**

Yes

**Related Work:**

2

**Strengths And Weaknesses:**

The overall position is clearly stated, however, I found several aspects of the supporting arguements to be unclear (see detailed comments/questions)

**Support:**

2

---

> ### Author Rebuttal · Authors · 2025-04-01
>
> We thank you for engaging deeply with our work! We appreciate your questions (Q), that helped us improve the clarity of our arguments, and for pointing out the typos in our document.
>
> (Q1) Drawback of co-existence: We agree that co-existence requires more effort and can be a slower process, as it is an open-ended and arbitrarily long-term interactive process. For example, it involves high levels of ambiguity (e.g., the exploration of a solution space without a set objective) and reflective engagement on behalf of the end user (e.g., the evaluation of said exploration).  We have expanded Section 6.4. with an in-depth discussion on the nature of the drawbacks of co-existing agents.
>
> (Q2) Relationship to humans: We would argue that all current embodied agents have clear reciprocal relationships with humans. In the example of the cleaning robot, the state that the human leaves the room is a form of interaction (e.g., not cleaning because you know the robot will do it, or arranging the room such that it can clean effectively). Even in fully autonomous setups, e.g., a factory floor, maintenance engineers are interactive, albeit on a longer and more asynchronous time scale. However, as discussed in Section 3.1, they currently lack the ability to sustain meaningful long-term interactions. We have expanded this discussion on the meaning of interaction in Section 1.
>
> (Q3) Double diamond: The double diamond is a fundamental concept in RtD that encapsulates the multiple convergent and divergent nature of the design process. We have placed Figure 2 in Page 6 and expanded the description of the double diamond in the introduction section.
>
> (Q4) Engineering and optimization: We apologize but due to lack of space, we refer you to (Q2) of reviewer “ygce”. We have clarified this point in Section 4.3.
>
> (Q5) Open-endedness in co-existence:  We argue that open-endedness permeates co-existence: it affects the ability of the agent to satisfy known objectives (e.g., using algorithms such as NEAT)  but also to participate (with the human and environment) in the emergence of new objectives. We have expanded this discussion in Section 6.1.
>
> (Q6) Generic nature: We define “generic” in the context of embodied agents as behaviors that are applied across many situations without accounting for the specific nuances of each situated environment or contextual diversity (as described in-depth in Section 2.2). To clarify the meaning of the term we have improved the description of the term in Section 2 and added a reference to Section 2.2. when the term is first mentioned.
>
> (Q7) Static Distribution: Thank you for the great question! We agree with the reviewer: as discussed in Section 6.3, we propose learning-based directions for handling non-stationary data distributions, while arguing that RL, continual and meta-learning are insufficient. Online RL can address non-stationarity (e.g. [2]–[4]), but typically relies on strong assumptions, such as knowing when and how dynamics change or using meta-learning priors, that limit its suitability for long-term co-existence. Moreover, although data distributions can evolve (as humans do), the current time-intensive deployment process means most agent behavior is learned pre-deployment. Thus, the agent’s embedded data distribution remains static. We expand on these limitations and related work (e.g. Lehman et al., 2025) in Appendix B.
>
> (Q8) Generic vs general: We agree we should clarify this difference beforehand. We have updated the beginning of Section 2 with this distinction between these two concepts.
>
> (Q9) Generalization in co-existence: Whilst we do think that the harmful stereotypes are in some cases a by-product of generalisation (often combined with data biases and hallucination properties), we don’t think agents should never generalise, as it is still a desirable feature for both autonomous systems and humans. However, under the current learning-based paradigm, achieving the nuanced care and attention that doctors demonstrate in the example of Section 2.1. through generalization would be unfeasible, as it would require the inclusion of increasingly fine grained physical and social cue data in the training process. In [5] the authors argue that this zero-shot generalization approach is exponentially expensive and therefore unfeasible. Instead, in this paper, we argue for agents that are able to continuously leverage the physical and social cues present in their specific environment to achieve co-existence. We have updated Section 2.2. to clarify that generalization does not always lead to harmful stereotypes but does have its limits in the context of co-existence.
>
> (Q10) Steamrolling: We apologize but due to lack of space we refer the reviewer to the weakness (W1) of reviewer “pjAt”. We have clarified this point in Section 2.3.
>
> [1] www.arxiv.org/abs/2211.06753
>
> [2] ​www.arxiv.org/abs/1905.03970
>
> [3] www.arxiv.org/abs/2102.05406
>
> [4] www.arxiv.org/abs/2105.08834
>
> [5] www.arxiv.org/abs/2404.04125

---

> > ### Comment · Reviewer_4ty4 · 2025-04-04
> >
> > I thank the authors for the detailed responses. I have updated my score accordingly.

---

### Decision · Program_Chairs · 2025-05-01

**Decision:**

Reject (with encouragement)

**Comment:**

Your paper was favorably reviewed, and we would have liked to accept it this year.  However, due to constraints on conference capacity, we had to reject some papers despite positive reviews.  The area chair's original meta-review (copied below) is a testament to the strengths of your paper.

----
The paper was reviewed by four expert reviewers.

The chief strengths of the paper include its clear position and interesting ideas. The paper clearly states its position on the co-existence of embodied artifical agents. The concept of ``co-existence'' of embodied artificial agents is somehow novel and interesting. The paper is well written by using compelling examples from biology to support its arguments. Also, the discussions on robust fail-safes, transparency, and accountability make the paper strong.

On the other hand, there were concerns about unclear supporting arguments and vague connections. For example, it was pointed out that the connection between design theory, evolution biology, and embodied agents were not straightforward. Also, there is a limited empirical evidence for supporting the ``steamrolling'' phenomenon. In addition, it was pointed out that the discussion on the framework developed from the co-existence concept is lacking.

The author-reviewer discussion has addressed most of the chief concerns about the paper. As a result, reviewers are generally positive about the apper. The AC agrees with the reviewers' opinions to reach this recommendation. The authors are encouraged to further revise the paper by clarifying the supporting arguments and address the potential drawbacks of the position.